# Pen drawing display

Sang-Mi Jeong[1,3], Taekyung Lim[1,3], Jeeyin Park[1], Chang-Yeol Han[2], Heesun Yang[2] & Sanghyun Ju [1]*

As advancements in science and technology, such as the Internet of things, smart home systems, and automobile displays, become increasingly embedded in daily life, there is a growing demand for displays with customized sizes and shapes. This study proposes a pen drawing display technology that can realize a boardless display in any form based on the user's preferences, without the usual restrictions of conventional frame manufacturing techniques. An advantage of the pen drawing method is that the entire complex fabrication process for the display is encapsulated in a pen. The display components, light-emitting layers, and electrodes are formed using felt-tip drawing pens that contain the required solutions and light-emitting materials. The morphology and thickness of each layer is manipulated by adjusting the drawing speed, number of drawing cycles, and substrate temperature. This study is expected to usher in the upcoming era of customized displays that can reflect individual user needs.

[1] Department of Physics, Kyonggi University, Suwon, Gyeonggi-Do 16227, Republic of Korea. [2] Department of Materials Science and Engineering, Hongik University, Seoul 04066, Republic of Korea. [3] These authors contributed equally: Sang-Mi Jeong, Taekyung Lim. *email: shju@kgu.ac.kr

Display devices have evolved significantly from early cathode-ray tube monitors to modern flat panel displays (e.g., liquid crystal displays, organic light-emitting diode (OLED) displays, and quantum dot LED (QD-LED) displays). Such evolving display devices eventually become commercial products available to general users in modern society. However, as current displays (such as computer monitors, smartphone displays, and navigation displays) are manufactured with a fixed rectangular shape, it is impossible to manipulate them to reflect the various design ideas of display users. Studies have been actively conducted on display fabrication methods based on printed assemblies of LEDs, QD (EML material) transfer, intaglio transfer, screen printing, and jet printing to overcome the constraints of rectangular-shaped displays[1–7]. Therefore, in recent years, the development of flexible, transparent, and holographic three-dimensional displays has been actively investigated for various applications[1,3,8–15].

Representative OLED and QD-LED displays usually consist of an anode, a hole injection layer (HIL), a hole transfer layer (HTL), an emission layer (EML), an electron transfer layer (ETL), an electron injection layer (EIL), and a cathode[16–18]. In general, thermal deposition and spin-coating methods are used to form light-emitting functional layers, such as the HIL, HTL, EML, ETL, and EIL, while thermal, electron-beam, and sputtering evaporation methods are used to form the anode and cathode. Although these methods provide reliable thin film quality with respect to luminance and current efficiency characteristics, they require a pixel patterning process based on a shadow mask or photolithography. Hence, these processes limit the display shapes that can be formed and the substrates that can be used.

By contrast, a solution-based pen-drawing method has the advantage of forming a free-style display pattern without being restricted to the conventional rectangular shape on the substrate. In addition, it allows fabrication on a flexible plastic or paper substrate as well as on a rigid glass substrate. Hence, it is more useful than the above-mentioned methods. Recently, studies on the fabrication of photovoltaics, transistors, and batteries using a brush pen have been reported[19–21]. However, the pen-drawing method with a brush pen is used for only one layer or several layers rather than for all the layers constituting the device; the remaining layers are formed using the conventional vacuum deposition process. In particular, fabricating a display device that requires two electrodes (anode and cathode) and multilayer thin film structures (HIL, HTL, EML, and ETL) using the pen-drawing method alone remains a challenge[21–28]. To the best of our knowledge, no study has reported the implementation of a display on paper using the pen-drawing method alone.

In this paper, we introduce a pen-drawing display that is fabricated using only drawing pens without thermal evaporation or spin coating. Specifically, we use six drawing pens containing two types of conducting solutions for the anode and cathode and four types of light-emitting functional solutions for the HIL, HTL, EML, and ETL. The changes in each layer's thickness and uniformity are investigated by manipulating the drawing speed, number of drawing cycles, and substrate temperature. In addition, the performance of the fabricated pen-drawing display is demonstrated on glass, plastic, and paper substrates.

## Results

**Fabrication procedure of pen-drawing display.** A schematic diagram of the display fabrication process using the pen-drawing method is shown in Fig. 1a. Eight solutions were prepared for the six-step pen-drawing display fabrication process: PH1000 (highly conductive PEDOT:PSS, 95 vol% PH1000, 5 vol% dimethyl sulfoxide (DMSO)) for the anode (step 1), PEDOT:PSS (100 vol% AI 4083) for the HIL (step 2), poly(N-vinylcarbazole) (PVK) for the HTL (step 3), three QDs (CdSeS@ZnS, CdZnSeS@ZnS, and CdZnS@ZnS QDs as red (R), green (G), and blue (B) emitters, respectively) for the EML (step 4), zinc oxide nanoparticles (ZnO NPs) for the ETL (step 5), and silver nanowires (Ag NWs) for the cathode (step 6).

The drawing speed was manipulated using a custom-built three-axis micropositioning system at room temperature (20 °C) (Fig. 1b). Compared with brush pens, pencils, fountain pens, and ball pens with various tip shapes, felt-tip pens facilitate the formation of a display with QDs and electrode solutions owing to their porous writing surface composed of pressed and layered fibers. Specifically, by varying the shape and width of a felt-tip pen, a free-style display shape with various light-emitting areas can be readily formed. In addition, such pens allow easy control of the morphology and width because the area touching the substrate during pen drawing can be maintained uniformly. Hence, we selected a felt-tip pen for our implementation. A felt-tip pen was first soaked in each solution. The solution was then transferred onto a substrate under handwriting pressure to form a wet film. The substrates used were Corning glass, polyethylene naphthalate (PEN) plastic film, and paper.

When the felt-tip pen in which each solution is absorbed is brought into contact with the substrate, a meniscus is formed at the solution–air interface by surface tension and capillary action. During writing, a shear force is applied on the solution between the two boundaries, namely, the solution–substrate and the solution–felt-tip pen interfaces (Fig. 1c)[22]. The shear force is dominated by the motion of the felt-tip pen. After formation of the meniscus at the solution–air interface and horizontal movement of the pen at a constant speed, the solution spreads along the substrate and a thin film is formed as the solvent evaporates. The morphology and thickness of the thin film are dependent on the drawing speed, number of drawing cycles, and substrate temperature. The drawing speed was uniformly maintained in the range of 10–30 mm/s using the custom-built three-axis micropositioning system controlled by a computer-programmable motion controller. It is necessary to optimize the number of drawing cycles, as it determines the uniformity of the films on the substrate. In particular, in the case of PH1000, which should be deposited first on the substrate as an anode layer, and the Ag NWs, which should be uniformly connected with each other as a cathode layer, the number of drawing cycles strongly affects the electrical conductivity and optical transmittance. In addition, the evaporation speed of the solvent remaining on the thin film formed by pen drawing was controlled by applying a temperature of 60 °C to the substrate so as not to damage the lower layer in the process of forming a specific layer using the pen-drawing method.

**Pen drawing display optimized by the drawing conditions.** The field emission scanning electron microscopy (FE-SEM) images in Fig. 2 show the relationship between the drawing speed and the thin film thickness of each layer (PH1000, PEDOT:PSS, PVK, QDs, ZnO NPs, and Ag NWs) after each pen-drawing step. Three drawing speeds of 10, 20, and 30 mm/s were selected, which are similar to the speed at which a person writes on average (~20 mm/s). In general, meniscus-guided deposition using the blade coating method or slot die coating method increases or decreases the film thickness depending on the coating speed[29,30]. If the coating speed is similar to or lower than the evaporation speed of the solvent (<0.1 mm/s), the film thickness decreases because the film dries while shear force is applied to the meniscus

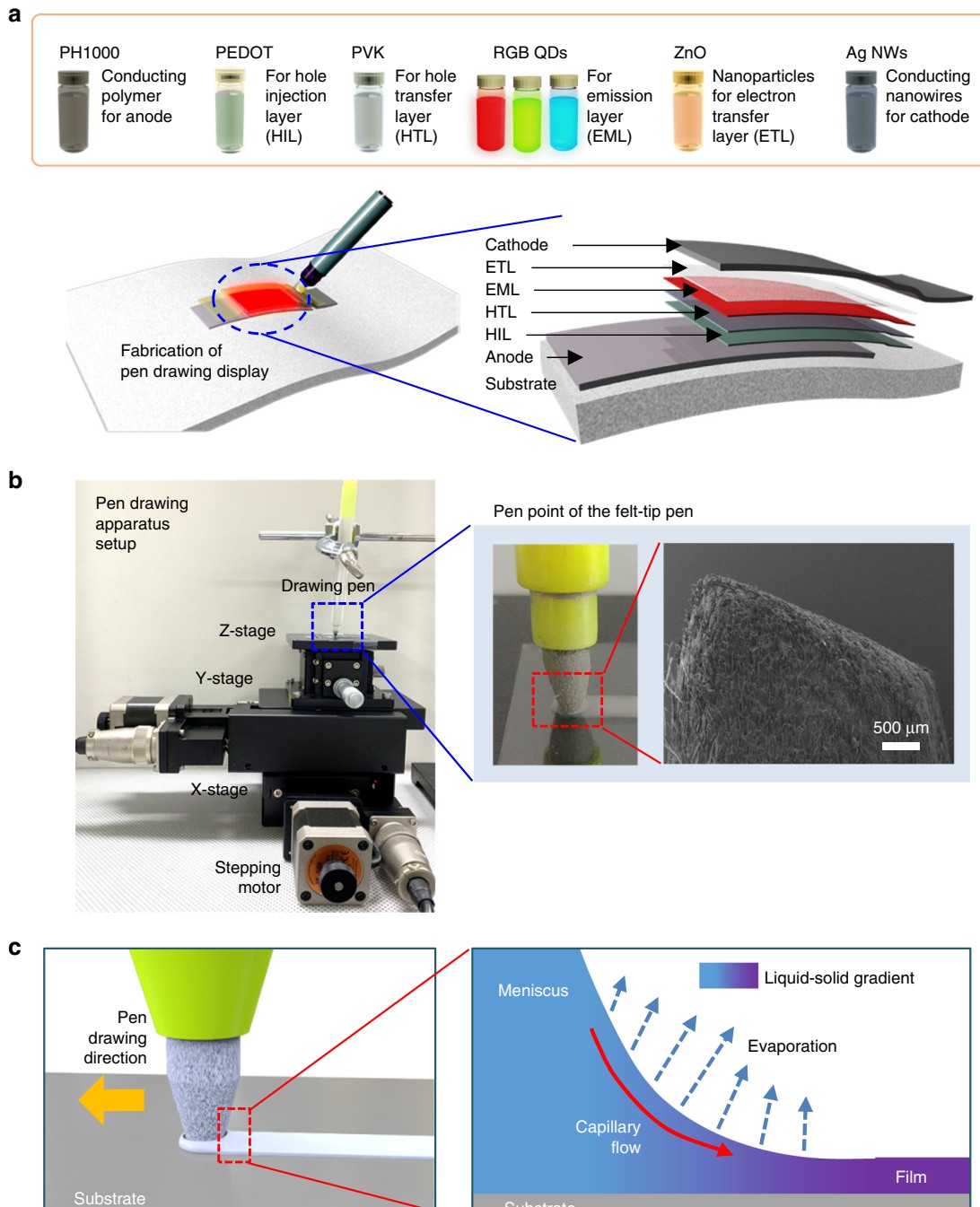

**Fig. 1** Pen-drawing display. **a** Schematic diagram of pen-drawing display with six drawing pens containing anode, hole injection layer (HIL), hole transfer layer (HTL), emission layer (EML), electron transfer layer (ETL), and cathode solutions. A quantum dot light-emitting diode (QD-LED) display fabricated using the pen-drawing method consists of PH1000 (anode), PEDOT:PSS (HIL), PVK (HTL), CdSeS@ZnS (red (R), EML), CdZnSeS@ZnS (green (G), EML), CdZnS@ZnS (blue (B), EML), zinc oxide nanoparticles (ZnO NPs, ETL), and silver nanowires (Ag NWs, cathode). **b** Optical image (left) of the pen-drawing apparatus setup. The pen drawing was performed using a custom-built three-axis micropositioning system controlled by a computer-programmable motion controller. Field emission scanning electron microscopy (FE-SEM) image (right) of the pen point of the felt-tip pen. **c** Schematic diagram and working principle of film formation through pen drawing

area between the substrate and the solution[31,32]. By contrast, in this study, if the coating speed for pen drawing is higher than the evaporation speed of the solvent, a wet film is formed outside the meniscus in accordance with the classical Landau–Levich regime. In this case, as the coating speed increases, the film thickness increases because a supersaturated wet film is formed by capillary flow in the meniscus area. The thickness ($h$) of the thin film can be expressed as a function of the capillary number

$C_a = \mu \cdot U / \sigma$, where $U$ is the coating speed and $\mu$ and $\sigma$ represent the viscosity and surface tension of the coating solution, respectively[33,34]. According to the equation $h = 1.34 \cdot C_a^{2/3} \cdot R_d$, the thickness of the thin film increases with the coating speed. Here, $R_d$ represents the radius of curvature of the downstream meniscus.

In the pen-drawing process at room temperature (20 °C), the solutions containing PH1000, PEDOT:PSS, PVK, QDs, ZnO NPs,

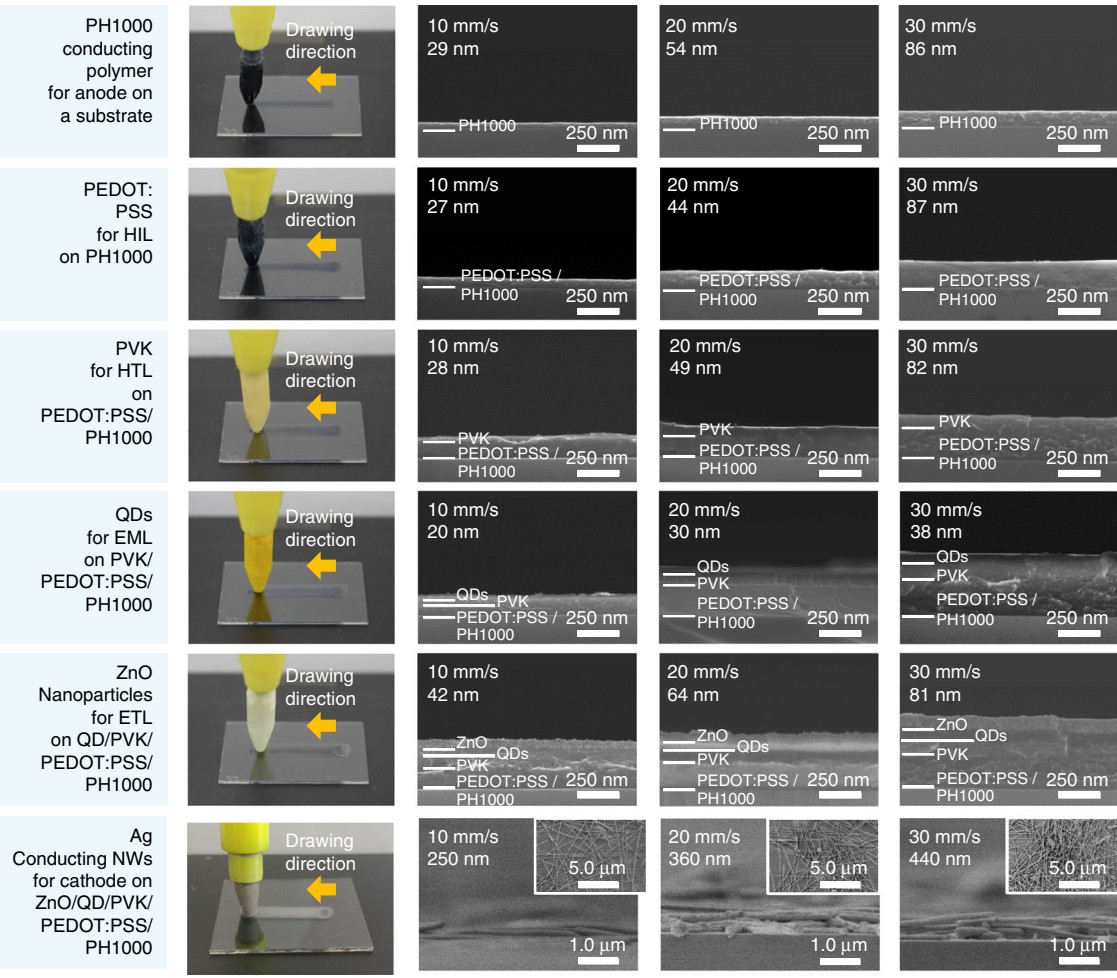

**Fig. 2** Gallery of pen-drawing images. PH1000, PEDOT:PSS, poly(*N*-vinylcarbazole) (PVK), QDs, ZnO NPs, and Ag NWs were deposited in order, and the thickness of each layer was observed using FE-SEM. The nanometer-scale thickness could be controlled by manipulating the drawing speed while the number of drawing cycles (1 half-cycle) and substrate temperature (20 °C) were fixed

and Ag NWs were dried as viscous wet films according to the classical Landau–Levich regime. Hence, the thickness increased with the increase in drawing speed from 10 to 30 mm/s (see Supplementary Fig. 1). For PH1000, PEDOT:PSS, and PVK, which are polymer-type materials, the viscosity increased rapidly as the polymer chains were entangled when the amount of solvent in the polymer solution decreased. As a result, the thickness of the polymer film increased exponentially with the drawing speed. This observation confirms that the thickness of the thin films is controlled in accordance with the classical Landau–Levich regime, where viscosity has a dominant effect. Unlike polymers, QDs, ZnO NPs, and Ag NWs have few physical interactions between particles, and the viscosity increased slightly with the concentration. Thus, the thickness increased linearly in proportion to the drawing speed.

Figure 3 shows the sheet resistance and transparency of the anode and cathode produced by the pen-drawing method. Owing to the structure of the QD display, for which multiple layers must be formed vertically, it is critical to form a uniform PH1000 film as the initial layer on the substrate. PH1000 is the most representative conductive complex in which PEDOT with a positive charge and PSS with a negative charge are mixed in a ratio of 1:2.5. The PEDOT:PSS complex in the PH1000 aqueous solution has a particular core–shell structure composed of hydrophobic PEDOT molecule cores surrounded by shells with hydrophilic PSS chains. Thus, for the PH1000 film formed by pen

drawing on a hydrophilic glass substrate at room temperature (20 °C), the hydrophobic and conductive PEDOT molecules are exposed to the surface as the moisture evaporates. The PEDOT molecules, which are hydrophobic, tend to aggregate owing to hydrophilic–hydrophobic repulsion with the glass substrate. PH1000 has a greater tendency for solution aggregation through hydrophilic–hydrophobic repulsion than the HIL (PEDOT:PSS layer (PEDOT:PSS ratio = 1:6)), HTL, EML, and ETL. Hence, it was difficult to obtain uniform and stable thin films based on the drawing speed alone. To solve this problem, we attempted to form a uniform PH1000 thin film by controlling the substrate temperature and the number of drawing cycles. Note that 1 cycle involves pen drawing in one direction for a certain area followed by the same operation in the opposite direction. To form a PH1000 thin film, we increased the substrate temperature (20, 40, 60, and 80 °C) under a fixed drawing speed (20 mm/s) and a fixed number of drawing cycles (1 cycle). As shown in the inset images in Supplementary Fig. 2b, when pen drawing was performed for 1 cycle, the PH1000 film was not formed uniformly in the pen-drawing area, even when the temperature was increased to 80 °C. However, when the transparency and sheet resistance were measured at the center and edge of the pen-drawing area, the differences in the transparency and sheet resistance of the PH1000 film at 60 °C were found to be smaller than those at the other temperatures (Supplementary Fig. 2a and b). Therefore, in this study, uniformity of the PH1000 film was

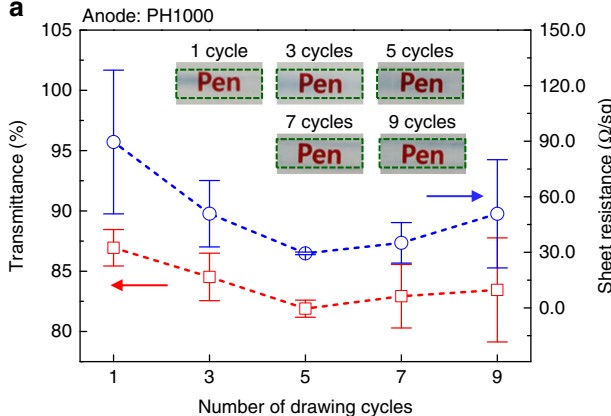

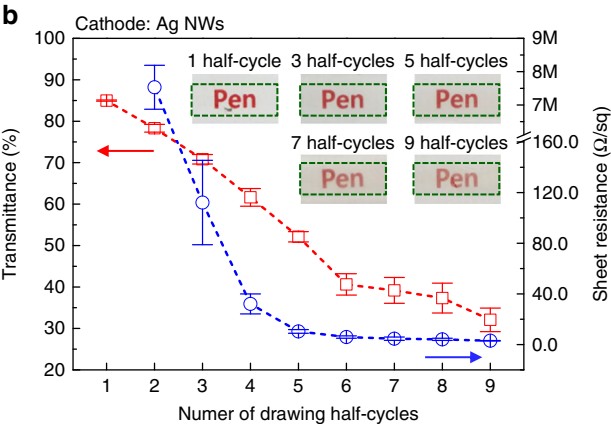

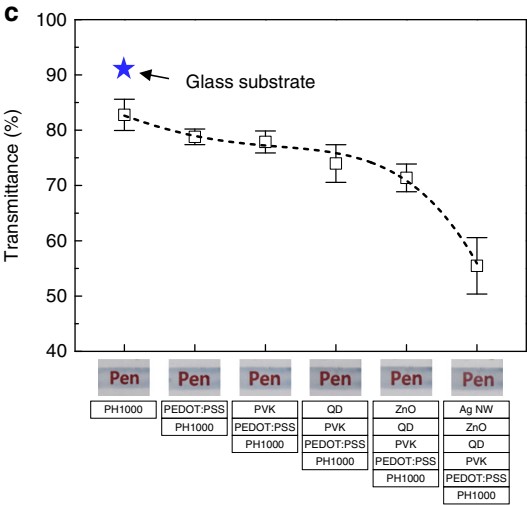

**Fig. 3** Electrical and optical properties of PH1000 and Ag NWs. Sheet resistance, optical transmittance, and photo images of PH1000 (**a**) and Ag NWs (**b**) depending on the number of drawing cycles. **c** Transmittance when each layer (PH1000, PEDOT:PSS, PVK, QDs, ZnO NPs, and Ag NWs) is sequentially formed on a glass substrate

achieved by adjusting the number of drawing cycles under a fixed substrate temperature of 60 °C. If the temperature is increased above 60 °C, a coffee-ring effect occurs owing to capillary flow at the edge of the pen-drawing area where the solvent evaporation speed is high, and additional PEDOT molecules are deposited at the edge (Supplementary Fig. 2d)[35].

Figure 3a shows the transmittance and sheet resistance of the PH1000 thin films formed by changing the number of drawing

cycles from 1 to 9 under a fixed substrate temperature of 60 °C. The red and blue open squares in this figure indicate the transmittance and sheet resistance of the PH1000 film that was pen-drawn on the glass substrate, respectively. When the number of drawing cycles increased from 1 to 3 and 5, both the transparency and the sheet resistance decreased and then slightly increased after 5 or more cycles. Furthermore, after 5 cycles of pen drawing, the error bar size notably decreased for both the transparency and sheet resistance data; thus, uniform transparency and sheet resistance values were observed. As shown in the inset images in Fig. 3a (enlarged images are shown in in Supplementary Fig. 3c), the PH1000 film appears thicker at the center after 1 and 3 cycles, whereas it appears thicker at the edge after 7 and 9 cycles. Furthermore, the transparency and sheet resistance measured at the center tended to decrease and then increase again after 5 cycles. By contrast, the transparency and sheet resistance of the edge tended to decrease gradually as the number of drawing cycles increased and then became constant (Supplementary Fig. 3). After 3 or less cycles of pen drawing, the PH1000 wet film formed at the edge dried extremely slowly and the hydrophobic properties of the PEDOT molecules were manifested. As a result, the PEDOT molecules showed a tendency to aggregate at the center of the PH1000 wet film formed on the hydrophilic glass substrate. Furthermore, after 7 or more cycles of pen drawing, the less dried solution at the center was applied to the previously dried film at the edge by the shear force. As a result, the film thickness at the edge increased while that at the center decreased (Supplementary Fig. 3). However, after 5 cycles of pen drawing, the solution was widely spread when the shear force was applied with the felt-tip pen until the PH1000 solution was dried and the viscosity increased. As a result, the PH1000 wet film was formed uniformly without aggregation at the center. Thus, a drawing speed of 20 mm/s, a substrate temperature of 60 °C, and 5 cycles of pen drawing were used as the pen-drawing conditions to form the optimal PH1000 layer.

The conductivity and transparency of the Ag NWs used as the cathode depend on the network state among the NWs. In other words, it is difficult to obtain uniform conductivity with just 1 half-cycle of pen drawing; if the number of pen drawing half-cycles increases, the conductivity increases but the transparency decreases. Thus, we attempted to determine the optimal conditions to achieve both conductivity and transparency. Note that 1 half-cycle involves pen drawing in only one direction for a certain area. If the duration of pen drawing for the Ag NWs (cathode layer) increases, the ZnO thin film (ETL), which is immediately below the cathode layer, can be dissolved by isopropyl alcohol (IPA), which is the solvent for the Ag NW solution. Hence, to prevent damage to the ZnO thin film, the substrate temperature was maintained at 60 °C to allow rapid drying of the pen-drawn Ag NW solution in IPA. Figure 3b shows the transparency and sheet resistance of the Ag NW layer controlled by the number of pen drawing half-cycles to connect the Ag NWs. It was verified that the transparency and sheet resistance decreased as the number of pen drawing half-cycles increased. In particular, a relatively stable sheet resistance of 150 Ω/sq or lower was observed after 3 or more pen drawing half-cycles.

From the experimental results presented above, the optimized pen-drawing conditions for each layer are as follows. Step 1: In the case of PH1000 for the anode, 5 cycles of pen drawing were performed at a drawing speed of 20 mm/s and a substrate temperature of 60 °C. Step 2: In the case of PEDOT:PSS for the HIL, 1 half-cycle of pen drawing was performed at a drawing speed of 20 mm/s and a substrate temperature of 60 °C. Step 3: In the case of PVK for the HTL, 1 half-cycle of pen drawing was performed at a drawing speed of 20 mm/s and a substrate

temperature of 20 °C. Step 4: In the case of the three QDs (CdSeS@ZnS, CdZnSeS@ZnS, and CdZnS@ZnS QDs as R, G, and B emitters, respectively) for the EML, 1 half-cycle of pen drawing was performed at a drawing speed of 20 mm/s and a substrate temperature of 20 °C. Step 5: In the case of the ZnO NPs for the ETL, 1 half-cycle of pen drawing was performed at a drawing speed of 20 mm/s and a substrate temperature of 20 °C. Step 6: In the case of the Ag NWs for the cathode, 3 half-cycles of pen drawing were performed at a drawing speed of 20 mm/s and a substrate temperature of 60 °C. The PEDOT:PSS layer was produced at 60 °C because, as with PH1000, the solvent for PEDOT:PSS is water and the drying speed is low if the layer is produced at room temperature (20 °C). In the case of the PEDOT: PSS layer, pen drawing was performed for only 1 half-cycle because a layer with uniform thickness was formed by increasing the substrate temperature to 60 °C. However, the solvents used for PVK, QDs, and ZnO are chlorobenzene, hexane, and ethanol, respectively, which are highly volatile. Thus, if the substrate temperature is increased, the thin film becomes rough owing to evaporation of the solvent. Therefore, for the PVK, QD, and ZnO layers, pen drawing was performed at room temperature (20 °C).

Figure 3c shows the transparency after each production step for the layers formed on a glass substrate with the optimized drawing speed, substrate temperature, and number of drawing cycles in ambient air. Thus, the transparency of the completed pen-drawing display produced on a glass substrate is ~55%. The anode and cathode showed decreases of approximately 10% and 15% in transparency, respectively, whereas the other light-emitting layers did not cause a significant change in transparency. Note that the pen-drawing display formed on the glass substrate by the pen-drawing method was transparent, such that the word "Pen" printed on paper could be seen clearly through the display.

For the pen-drawing display composed of six multilayers, the morphology of each layer has a significant effect on the deposition of the next layer. In other words, the rougher the lower layer, the rougher is the layer deposited above it. Moreover, as each layer except the Ag NW layer is a thin film with a thickness of several tens of nanometers, if the uniformity of each film decreases, a short may occur in the thin part during display operation. Thus, each layer must be deposited uniformly when a multilayered display device is produced. Figure 4 shows the atomic force microscopy (AFM) image of each layer formed by pen drawing. Each layer was formed using the handwriting method at a drawing speed of ~20 mm/s. Furthermore, the optimized substrate temperature and number of drawing cycles, as shown in Fig. 3, were used. The surface morphology of each layer formed by the pen-drawing method was compared with that of the corresponding layer formed by the spin-coating method (Supplementary Fig. 4). Consequently, the root-mean-square (RMS) roughness ($R_{rms}$) values of the PH1000, PEDOT:PSS, PVK, QD (G), ZnO, and Ag NW layers were found to be 1.609, 1.245, 0.700, 1.177, 1.271, and 110.120 nm, respectively, for pen drawing and 1.278, 1.158, 0.600, 1.456, 1.423, and 90.114 nm, respectively, for spin coating. To investigate the relationship between the width of the felt-tip pen and the line width formed on a substrate, pen drawing of PH1000 films was performed using felt-tip pens with various widths, which dictated the line shape of the obtained film (Supplementary Fig. 5). These results indicate that pen-drawing produces smooth and uniform thin films that can be used as displays.

**Light-emitting characteristics of the pen-drawing display.** Figures 5a–c show the band diagrams of the pen-drawn QD-LEDs consisting of PH1000/PEDOT:PSS/PVK/QD (R, G, and B)/ ZnO NP/Ag NW structures. Note that all the layers were formed

using the pen-drawing method. The measured work function values of the PH1000 and Ag NWs were 5.0 and 4.8 eV, respectively. PVK, which is an organic material used as the HTL, has a deeper highest occupied molecular orbital (HOMO) level (−5.4 eV) and shallower lowest unoccupied molecular orbital (LUMO) level (−1.9 eV) than other HTL materials. Hence, the HOMO level of PVK lies between the valence band maximum (−6.8 to −7.2 eV) of the QDs, which are the EML, and the work function (−5.0 eV) of the anode. Furthermore, the large potential energy gap between the LUMO of PVK and the conduction band minimum (−4.4 to −4.8 eV) of the QDs plays a crucial role in preventing electron transfer from the QD EML to the HTL, which may be caused by the electric field[36]. The electron affinity and ionization potential of the ZnO NPs were ~4.3 and ~7.6 eV, respectively. The ZnO NPs as the ETL not only facilitate the injection of electrons into the QDs but also block hole transport at the QD/ETL interface and consequently provide increased exciton recombination efficiency in the QD layer region.

Figures 5d–f show the luminance–current–voltage (LIV) characteristics of the pen-drawn QD-LEDs, which showed R, G, and B emissions. The pen-drawing display device consists of PH1000 (~100 nm) as the anode, PEDOT:PSS (~40 nm) as the HIL, PVK (~50 nm) as the HTL, QDs (CdSeS@ZnS for R/CdZnSeS@ZnS for G/CdZnS@ZnS for B; ~30 nm) as the EML, ZnO NPs (~60 nm) as the ETL, and Ag NWs (~600 nm) as the cathode. The Ag NW layer as the cathode was relatively thick compared with the other deposited layers, forming an NW network with high conductivity. The luminescence characteristics of the pen-drawn QD-LEDs showed turn-on voltages of ~7 (R), ~7 (G), and ~9 (B) V and maximum luminance values of 125 (R), 72 (G), and 5.6 (B) cd/m$^2$ at 9.4, 9.25, and 10.4 V, respectively. The current efficiencies were 0.031 (R), 0.020 (G), and 0.004 (B) cd/A at 9.3, 9.25, and 10 V, respectively.

The poor luminance and efficiency of the fabricated devices were mainly due to the high resistivity of the electrodes, relatively poor contact resistance, poor transmittance, and the lack of reflection from the transparent metallic cathode. A high luminance with a low turn-on voltage is dependent on the sheet resistance of the electrodes. Supplementary Fig. 6 shows the luminance of a display produced by depositing the HIL, HTL, EML, and ETL by pen drawing and depositing the anode and cathode layers by sputtering with indium tin oxide (ITO; ~100 nm) and aluminum (Al, ~150 nm), respectively. Compared with the sheet resistances of ~8.7 Ω/sq for ITO and ~1.3 Ω/sq for Al, the sheet resistances of the pen-drawn PH1000 (~30 Ω/sq) and Ag NWs (~110 Ω/sq) as anode and cathode, respectively, were considerably higher. This difference in the resistance characteristics seems to have significantly affected the luminance and efficiency. Furthermore, the characteristics also depend on whether the luminance is measured at the anode or cathode of the produced device. For conventional light-emitting devices, the luminance is measured in the direction of the transparent anode (ITO). Thus, the measured luminescence and efficiency increase because the light emitted from the LED is reflected by the opaque and glassy metallic cathode (Al), and it progresses in the anode direction. However, for the LED produced by pen drawing, there is no opaque and glassy material such as Al in the lower film, and only some of the light emitted by the Ag NW layer as the cathode with 70% transparency is reflected. As a result, the measured luminance and efficiency of the pen-drawing display in which the anode and cathode were formed by the pen-drawing method were lower than those of the pen-drawing display with forming the anode and cathode by the conventional sputtering process. Thus, such issues should be further investigated to fabricate brighter and more efficient pen-drawn QD-LEDs with novel conducting electrode materials. As shown in

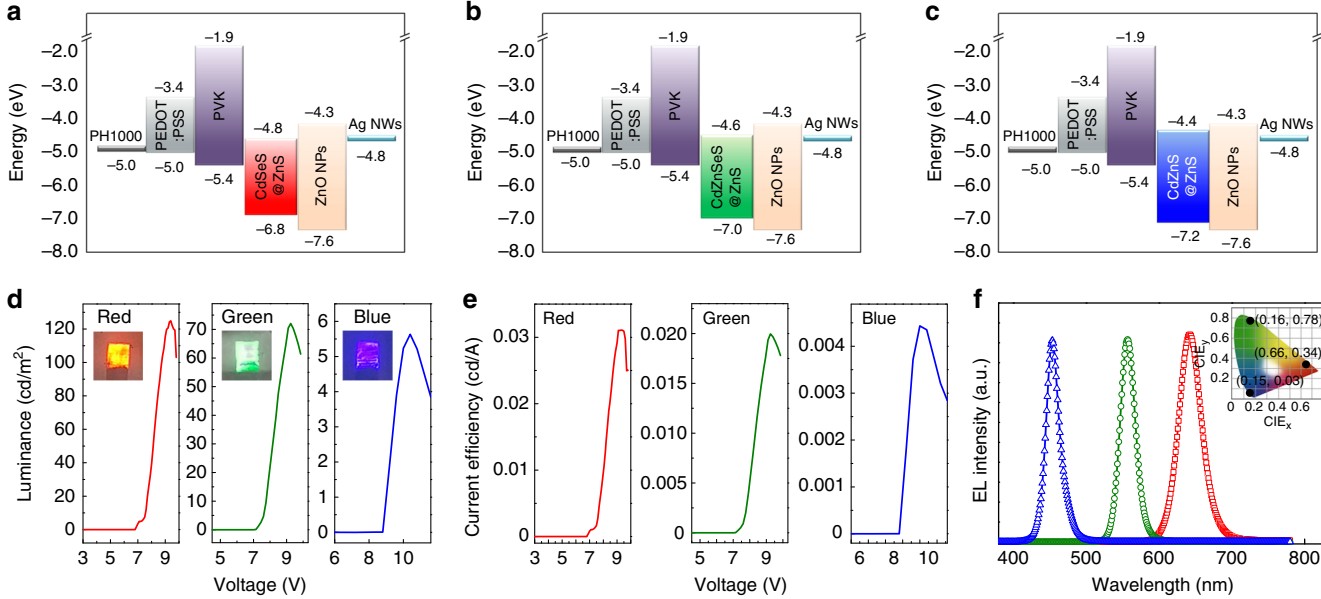

**Fig. 4** Surface morphology of each layer formed by pen drawing. Atomic force microscopy (AFM) topographic images, height section analyses, and three-dimensional images of PH1000 (**a**), PEDOT:PSS (**b**), PVK (**c**), QDs for G (**d**), ZnO (**e**), and Ag NWs (**f**)

**Fig. 5** Luminance–current–voltage (LIV) characteristics of the QD light-emitting diodes (QD-LEDs) fabricated using the pen-drawing method. Energy levels of red (R), green (G), and blue (B) QD-LEDs. Highest occupied molecular orbital (HOMO) and lowest unoccupied molecular orbital (LUMO) levels of PH1000, PEDOT:PSS, PVK, QDs, ZnO NPs, and Ag NWs. CdSeS@ZnS for R emission (**a**); CdZnSeS@ZnS for G emission (**b**); and CdZnS@ZnS for B emission (**c**). Luminance (**d**), efficiency (**e**), and electroluminescence spectra and the Commission Internationale de l'Elcairage (CIE) color coordinates (**f**) corresponding to the R, G, and B emitting QD-LEDs with the structure of PH1000/PEDOT:PSS/PVK/QDs (CdSeS@ZnS for R, CdZnSeS@ZnS for G, and CdZnS@ZnS for B)/ZnO NPs/Ag NWs

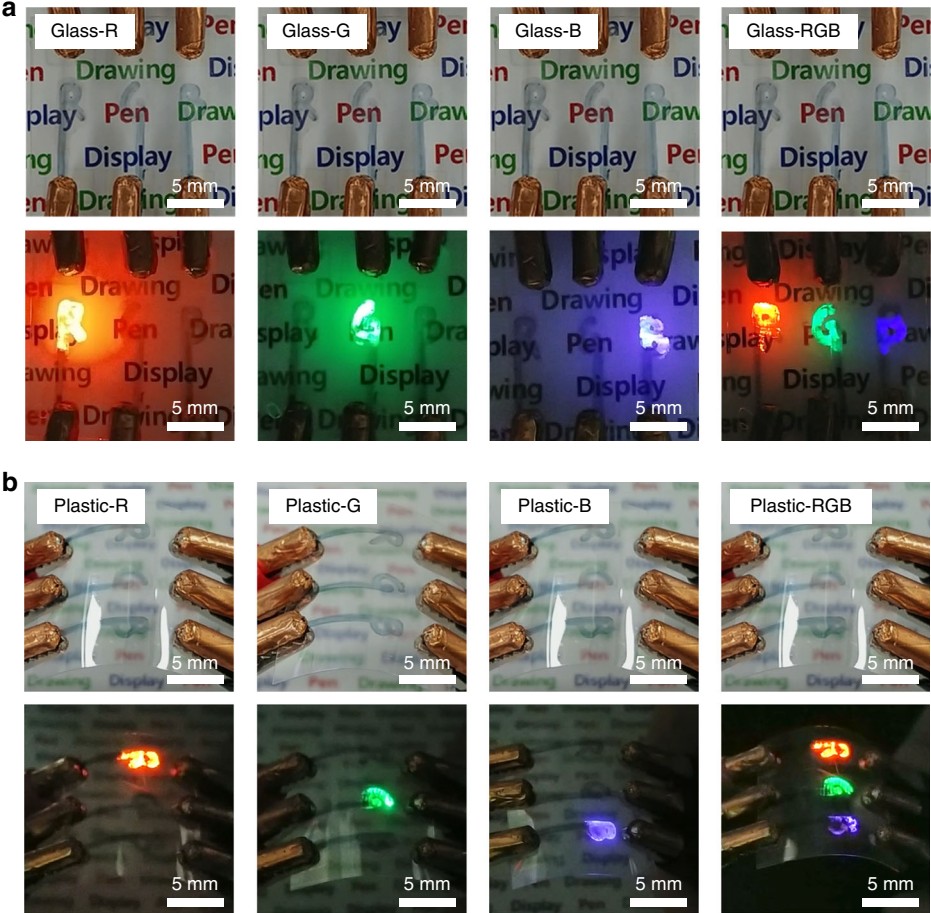

**Fig. 6** Handwriting display images using the pen-drawing method. Photo images of "RGB" text displays formed on the glass (**a**) and plastic (**b**) substrates, showing the optical clarity and mechanical flexibility

Fig. 5f, the electroluminescence spectrum of the pen-drawn QD-LEDs had maximum peaks of 643 (R), 557 (G), and 455 nm (B) with remarkably narrow full-width-at-half-maximum values of 38 (R), 25 (G), and 24 nm (B). Thus, the fabricated device can afford exceptional color purity. As indicated in the inset of Fig. 5f, the pen-drawn QD-LEDs showed the Commission Internationale de l'Elcairage (CIE) color coordinates of (0.66, 0.34) (R), (0.16, 0.78) (G), and (0.15, 0.03) (B).

**Implementation of the pen-drawing display**. Figure 6 shows free-style multicolor text "RGB" drawn on various transparent, flat, and flexible substrates, such as glass and plastic substrates, using the pen-drawing method. Note that the RGB patterns were formed by hand while maintaining the drawing speed, number of drawing cycles, and substrate temperature. Figure 6a shows the display formed by hand drawing the RGB text on a glass substrate, with the letters R, G, and B drawn from the anode to the HIL, HTL, EML, ETL, and cathode. Figure 6b shows the display formed on a plastic substrate in a similar manner. Even though the plastic substrate is bent, the hand-drawn RGB letters are emitted clearly. The advantage of the pen-drawing method for display fabrication is that displays can be formed in any shape.

Figure 7 shows the process for producing a pen-drawing display on paper and light-emitting RGB images of the fabricated device. To produce the pen-drawing display on paper, eight solutions (PH1000, PEDOT:PSS, PVK, R QDs, G QDs, and B QDs, ZnO NPs, and Ag NWs), eight felt-tip pens, and paper substrates were prepared. Ordinary paper cannot be used because

its surface is too rough and it becomes warped when it is wetted by the solutions. Hence, to ensure minimum paper roughness, we used photo paper that is often used in general inkjet printers (260 g/m$^2$; thickness, 0.25 mm; Formtec photo paper, IH3057). Furthermore, a 300 nm-thick silicon dioxide (SiO$_2$) thin film was deposited on the paper substrate to prevent the solution from permeating into the cellulose fibers and making the thin film thinner or irregular. As shown in Fig. 7, the pen-drawing display was produced by sequentially forming layers of PH1000, PEDOT: PSS, PVK, QDs, ZnO NPs, and Ag NWs. In particular, in step 4, the background was drawn with B and G QDs and the text "Display" was written with R QDs on the background by using a thin felt-tip pen with a pen width of 0.4 mm. When the text formed by pen drawing was observed under ultraviolet (UV) light irradiation, the RGB text appeared vividly and some overlap was observed between the pen-drawn RGB areas. The images at the bottom of Fig. 7 shows the produced pen-drawing display under UV light irradiation and various conditions, namely, application of a voltage of 14 V, bending of the paper substrate, and cutting of the paper substrate (from the right). All of the images vividly show the "Display" text. In the area where two QDs overlap, a mixture of the two colors can be observed. As shown in Fig. 5, as the luminance decreases in the order of R, G, and B, the "Display" text is illuminated vividly even though the text was drawn with R QDs on top of the areas in which G and B QDs were drawn. Thus, by using the pen-drawing method proposed in this study, various texts and figures can be handwritten on flexible substrates (Supplementary Fig. 7). Various text colors can be created easily by mixing the RGB QDs, and it is expected that figures with

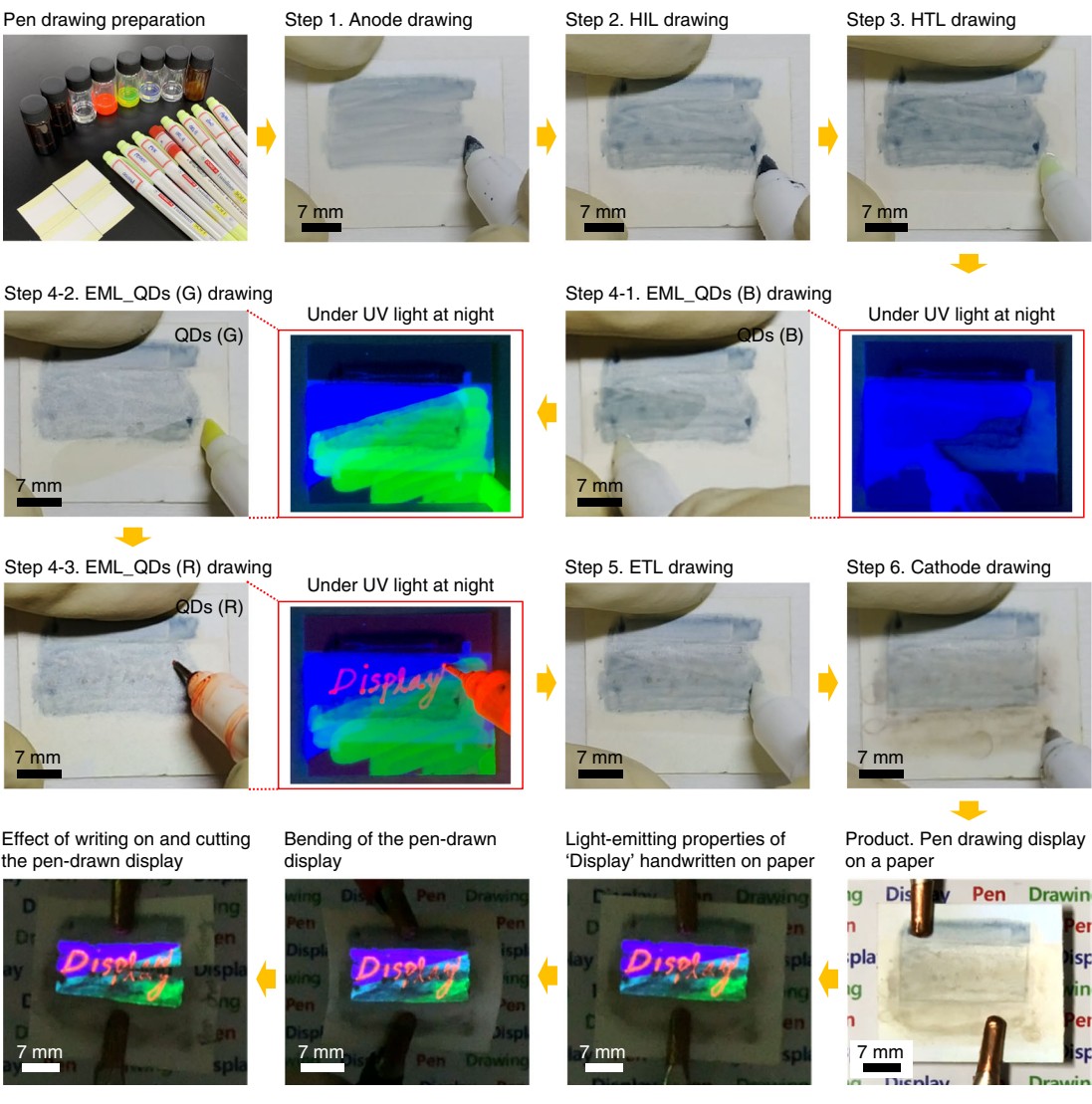

**Fig. 7** Handwriting images using the pen-drawing method. Photo images of "Display" handwritten on paper, showing the light-emitting ability, mechanical flexibility, and tearing properties

various textures can be obtained by using felt-tip pens with assorted thicknesses.

## Discussion

In summary, a pen-drawing method was introduced to fabricate QD-LED displays that could be formed on substrates of various shapes and materials in a free-style manner. The pen-drawing display could provide full-color light-emitting properties. The thicknesses of the pen-drawn materials, such as the electrode layers (anode and cathode) and light-emitting layers (HIL, HTL, EML, and ETL), could be controlled by adjusting the drawing speed, number of drawing cycles, and substrate temperature. Each layer could be safely formed on the deposited underlying layers without causing any damage to them. The main advantage of pen-drawing displays is that they can be formed not only on flat substrates (e.g., wafers and glasses) and flexible substrates (e.g., plastics and papers) but also on various complex substrates such as concave–convex structures (e.g., cups, walls, and pillars) based on the user's free-style design.

## Methods

**Preparation of conducting solutions for anode and cathode**. PH1000 (Clevios), which is a mixture of PEDOT:PSS (1:2.5 by weight) in aqueous solution, was used

as the conducting solution for the anode. PH1000 had a solid content of 1.0–1.3%, a viscosity of 15–50 mPa·s, and a conductivity of ~850 S/cm. To increase the conductivity, PH1000 and 5 wt% DMSO (99.9% ACS reagent, Sigma-Aldrich) were mixed and stirred for 24 h. The conducting solution for the cathode consisted of 0.5 wt% Ag NWs (Sigma-Aldrich) in IPA. The average diameter and length of the Ag NWs were ~60 nm and ~10 μm, respectively.

**Preparation of the solutions for PEDOT:PSS, PVK, QDs, and ZnO NPs**. PEDOT:PSS (Clevios, AI 4083), which is a mixture of PEDOT:PSS (1:6 by weight) in aqueous solution, was used for the HIL layer. The PEDOT:PSS solution had a solid content of 1.3–1.7%, a viscosity of 5–12 mPa·s, and a resistivity of 500–5000 $\Omega$·cm. The PVK solution for the HTL was prepared by dissolving 0.05 g of PVK (average MW = 25,000–50,000) in 5 mL of chlorobenzene. Dispersions of CdSeS@ZnS (R), CdZnSeS@ZnS (G), and CdZnS@ZnS (B) QDs in hexane with concentrations of ~43, ~20, and 23 mg/mL, respectively, were used for the EML, and a dispersion of ZnO NPs in ethanol with a concentration of ~30 mg/mL was used for the ETL.

**Fabrication of pen-drawing displays**. Pen drawing displays were fabricated on Corning glass 1737, PEN plastic film (1.36 g/cm³; thickness, 0.125 mm; Teijin DuPont Films, Teonex® Q65HA), and photo paper (260 g/m²; thickness, 0.25 mm; Formtec photo paper, IH3057) substrates using two types of felt-tip pens for bold text (width, 4.0 mm; DONG-A, Twinliner SOFT) and thin text (width, 0.4 mm; Monami, namepen·x). Each felt-tip pen was emptied of all existing ink and soni-cated for 24 h in IPA to remove residual ink; the solvent containing the pen was changed frequently until it became colorless. Subsequently, the felt-tip pen was sonicated in DI water for 24 h and then dried naturally for 24 h.

For the paper substrate used in this study, to prevent the solution from penetrating into the cellulose fibers of the paper (which could result in random film thicknesses), a 300 nm layer of $SiO_2$ was deposited by sputtering on the paper. To obtain films of uniform thickness by preventing the agglomeration of the PH1000 solution, pen drawing of the PH1000 layer was performed for 5 cycles. In addition, to achieve conductivity and transparency while reducing the damage to the lower layer, pen drawing with the Ag NW solution was performed for 3 half-cycles. Here, 1 half-cycle involves pen drawing in only one direction for a certain area, and 1 cycle involves 2 half-cycles of pen drawing in opposite directions. For the four light-emitting layers (PEDOT:PSS, PVK, QDs, ZnO NPs), only 1 half-cycle of pen drawing was performed

The felt tips of the drawing pens, which contained adequate amounts of each solution (PH1000, PEDOT:PSS, PVK, QDs, ZnO NPs, and Ag NWs), were lowered to touch the substrate, and while the felt-tip pen was horizontally moved on the substrate, a thin film was formed, which gradually dried with time from the starting point of pen drawing. Pen drawing display patterns were produced using a mechanical drawing apparatus (Fig. 1b) through the following procedure (Fig. 7). Step 1: First, a PH1000 thin film as the anode layer was drawn with a drawing speed of ~20 mm/s, 5 drawing cycles, and a substrate temperature of ~60 °C. It was then baked at 150 °C for 30 min The deposited thickness of the PH1000 thin film was ~100 nm. Step 2: An ~40-nm thick layer of PEDOT:PSS as the HIL was deposited on top of the PH1000 thin film via pen drawing with a drawing speed of ~20 mm/s, 1 half-cycle, and a substrate temperature of 20 °C. It was then baked at 150 °C for 30 min. Step 3: A PVK layer was formed with a thickness of ~50 nm as the HTL on top of the HIL with a drawing speed of ~20 mm/s, 1 half-cycle, and a substrate temperature of 20 °C. It was then baked at 150 °C for 30 min. Step 4: R, G, and B QD dispersions were used to deposit the EML with a drawing speed of ~20 mm/s, 1 half-cycle, and a substrate temperature of 20 °C, followed by baking at 60 °C for 30 min The deposited thickness of the QD thin film was ~30 nm. Step 5: A ZnO NP thin film as the ETL with a thickness of ~60 nm was drawn with a drawing speed of ~20 mm/s, 1 half-cycle, and a substrate temperature of 20 °C, followed by baking at 60 °C for 30 min. Step 6: Finally, Ag NWs (~600 nm thick) as the cathode layer were deposited with a drawing speed of ~20 mm/s, 3 half-cycles, and a substrate temperature of 60 °C on top of ETL, followed by baking at 60 °C for 30 min. As a result, the pen-drawn QD-LED consisted of a multilayered structure of PH1000/PEDOT:PSS/PVK/RGB QDs/ZnO NPs/Ag NWs.

**Device characterization**. The thicknesses and surface morphologies of the individual layers (anode, HIL, HTL, EML, ETL, and cathode) were observed using FE-SEM (S-4800, HITACHI). The sheet resistances of the PH1000 and Ag NW electrodes were measured using a four-point probe system (CMT-SR1000N, AIT Co., Ltd.). The optical transmittance of the pen-drawn QD-LED devices was measured using an ultraviolet–visible–near-infrared (UV–vis–NIR) spectrometer (UniART, UniNano Tech.). The optical characteristics, in terms of luminescence, efficiency, emission spectrum, and CIE color coordinates, of the pen-drawn QD-LEDs were measured using an LIV measurement system with a Konica Minolta spectroradiometer (CS 2000) coupled to a Keithley voltage and current source (K2400).

## Data availability

The data that support the plots within this paper and other findings of this study are available from the corresponding author upon reasonable request.

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

## Acknowledgements

This work was supported by the National Research Foundation of Korea (NRF) grant funded by the Korea government (MSIP) (2017M3C1A9069593,

2017R1D1A1B04030415, 2017R1A2B3008628, 2018M3A7B4070987, 2019R1C1C1010688, and 2019R1A2C2010614).

## Author contributions

S.J. conceived and designed the study. S.-M.J., T.L., and J.P. carried out experiments. C.-Y.H. and H.Y. provided QD materials. S.J. wrote the paper. All authors contributed to discussions regarding the research.

## Competing interests

The authors declare no competing interests.
