## [Peer Review File · Nature Communications]

Reviewers' comments:

Reviewer #1 (Remarks to the Author):

The manuscript from Ju et al. describes the fabrication of LEDs and displays by direct pen writing of conductive and fluorescent inks. Although pen writing of these materials has been shown, the great novelty of this work is the demonstrated ability to write in a layer-by-layer fashion in order to generate functional devices such as displays. The authors described in great detail the variation in experimental conditions leading to optimum homogenous deposition of components one on top of the other. Although, as the authors themselves pointed out, still some improvement could be achieved in the performance of the final assembled LEDs, the results reported are novel and encouraging for the development of flexible displays.

I would therefore recommend this manuscript for publication after authors comment on the following points: 1) The direct pen writing showed here is simple and low cost and can achieve great control of material deposition. However, I'm not sure the method would be able to compete with other existing techniques for the generation of large area displays. 2) Have the authors tested other RGB materials alternative to QDs?. Do they see any obstacles in use of polymers or nanostructured polymer materials?. 3) Were the pens used empty of ink or were they emptied and washed prior use?

Reviewer #2 (Remarks to the Author):

SUMMARY & ANALYSIS

The paper describes the first hand-drawn, color, electroluminescent, quantum dot (QD) display. Prior work has reduced the complexity of QD display fabrication from a cleanroom process, to a printed process (especially Ref. 4). This work further reduces the barrier to entry for QD display fabrication by replacing printing with a hand-drawn process while preserving and extending the ability to fabricate on a wide variety of stiff and flexible substrates. The results are good, considering the fabrication method. The artifacts adequately establish the feasibility of hand-drawn quantum dot displays, especially Figure 7.

COMMENTS

Abstract:

What is the significance of 'boardless' in this context? How is this display boardless when it requires a substrate (e.g. glass and plastic per Figure 6)?

As a primary motivation, diversifying display form factors seems weak without an example to separate it from the previous printing work of Ref 4.

What are 'frame' manufacturing techniques?

Introduction:

How do flexible, transparent and holographic displays address the problem of fixed rectangular image shape?

Page 3

The claim that this is the first fully pen-drawn display is both significant and novel for QD displays. The creation of a display that is constructed entirely on a bench and on a wide variety of substrates has been previously reported both for QDs and on other electroluminescent platforms. EL (TFEL) is an example of a display technology that can be screen-printed, with multiple colors, on a variety of materials (wood, plastic, leather). See reference below:

Olberding, Simon, Michael Wessely, and Jürgen Steimle. "PrintScreen: fabricating highly customizable thin-film touch-displays." Proceedings of the 27th annual ACM symposium on User interface software and technology. ACM, 2014.

Perhaps the authors should acknowledge TFEL and screen-printed displays. QDs may have a number of significant advantages (lower drive voltage, higher efficiency) but it seems appropriate to acknowledge these displays as they have shared many of the goals set out in this paper and

they are present in many of the target applications mentioned in the introduction.

Page 5

How sensitive are the display characteristics to the parameters listed here such as pen speed, material thickness, and uniformity? Can we reasonably expect a hand-drawn display to produce results similar to the carefully controlled machine-drawn results described? Was the work in Figure 7 the result of multiple attempts with these parameters varied? I feel that this may warrant some discussion as it is the contribution of this paper which diverges most significantly from previous work.

Page 14

The first sentence of the last paragraph is not very precise. What is meant by the words, 'actual device.'

References:

The references could include TFEL as noted above.

Figures:

The figures are effective artifacts and the photos appear to be free of inappropriate modification.

Reviewer #3 (Remarks to the Author):

This study offers an interesting inexpensive approach to customizing the formation of simple devices using pen drawing. The process and concept are minimally described - a number of general statements and claims are made that need justification and in some cases, an indication of values and the variance observed or expected.

Lines 79-81: What is the range of line widths that can be readily generated? Does this require pens with different tips? What is the variance or reproducibility in line width for each tip? How does this affect device properties such as luminance of the QD-LEDs?

Lines 99-102: What is a "suitable temperature"? Film properties will vary with heating temperature and probably heating time, so this statement gives no useful information.

Lines 151-154: The process apparently balances convective flow and capillary flow. How controllable and reproducible is this effect? The terms uniformity (line 155) and uniform (line 166) are used, but these terms have no meaning. How reproducible are the values of sheet resistance and transparency? Does this depend upon the air flow rate, humidity, room temperature, etc.? What variance is observed?

Lines 283-285: What does "somewhat lower" mean?

Lines 362-366: After depositing 300 nm of SiO₂ onto paper, the surface is essentially that of glass. What is the purpose of using paper substrates in this situation (I presume it is a cost issue, but the deposition adds cost also). Does the paper maintain its flexibility after deposition? Does the glass crack upon bending? Is the substrate curved after film deposition due to differences in thermal expansion coefficient?

Nearly all figures in the manuscript and in Supplementary Information will be very difficult to read when reduced to fit in journal columns. Text and numbers must be MUCH larger, especially on Figs. 2-6 and Figs. S2-S6.

Response Letter

We appreciate the reviewers' comments on our manuscript, which we found helpful in improving the clarity of our presentation. We have made revisions in response to all the specific comments, and are submitting the revised manuscript (enclosed) for further consideration for publication in *Nature Communications*.

Reviewer #1 (Remarks to the Author):

The manuscript from Ju et al. describes the fabrication of LEDs and displays by direct pen writing of conductive and fluorescent inks. Although pen writing of these materials has been shown, the great novelty of this work is the demonstrated ability to write in a layer-by-layer fashion in order to generate functional devices such as displays. The authors described in great detail the variation in experimental conditions leading to optimum homogenous deposition of components one on top of the other. Although, as the authors themselves pointed out, still some improvement could be achieved in the performance of the final assembled LEDs, the results reported are novel and encouraging for the development of flexible displays. I would therefore recommend this manuscript for publication after authors comment on the following points:

Comments 1. The direct pen writing showed here is simple and low cost and can achieve great control of material deposition. However, I'm not sure the method would be able to compete with other existing techniques for the generation of large area displays.

Response: We appreciate the reviewer's comment. An advantage of the pen drawing display is that the entire complex fabrication process for the display is encapsulated in a pen. Therefore, it is possible to implement a display having various color patterns by using a display pen only, without requiring conventional complex display fabrication processes such as deposition, alignment, patterning, and transferring. The main goal of the pen drawing display is to increase the applicability of displays through a differentiated fabrication method, not to replace conventional commercialized large displays (LCD, OLED, and QD-LED).

The differentiation and applicability of the pen drawing display are as follows. (i) It is possible to fabricate a display by only using a pen. Thus, displays can be produced easily and rapidly at a desired location. (ii) There is no limitation on the substrate (e.g., paper, plastic, and glass) or the surface morphology of the product (curved, large area, or complex structure); thus, it is easy to produce indoor and outdoor displays/illuminations/signboards for building walls, windows, department store pillars, offices, and vehicles as well as for places where it is difficult to install conventional displays, such as mountains and islands, or places where signboards must be replaced frequently. As a typical example, a user could easily display an advertisement consisting of various text characters or image shapes without being affected by the size or curvature of the glass window of a store.

Comments 2. Have the authors tested other RGB materials alternative to QDs? Do they see any obstacles in use of polymers or nanostructured polymer materials?

Response: In this study, the QD-LEDs fabricated using the pen drawing method were composed of six layers (PH1000, PEDOT:PSS, PVK, QDs, ZnO NPs, and Ag NWs). The PH1000, PEDOT:PSS, and PVK layers are made of polymer-type materials; the QDs and ZnO NPs are small particles; and the Ag NWs are long wire-shaped nanostructures. It should be noted that for the pen drawing process, PH1000 (Clevios) and PEDOT:PSS (Clevios, AI 4083) aqueous solutions, which are commercially available reagents, and PVK, QDs, ZnO NPs, and Ag NWs were dispersed in chlorobenzene, hexane, ethanol, and IPA, respectively (please see the Methods section). As shown in the AFM images in Fig. R1, the surface morphologies of each layer formed by the pen drawing method are similar to those formed by a spin-coating method.

Based on the correlation between thickness and drawing speed for each material (Fig. R2), in the case of the polymer-type materials (PH1000, PEDOT:PSS, and PVK), the thickness of the polymer film increased exponentially with the drawing speed. Unlike the polymers, because the QDs, ZnO NPs, and Ag NWs have few physical interactions between particles, the thickness increased in linear proportion to the drawing speed. Therefore, for the materials used in this study, such as polymers and nanoparticles as well as QDs, it was confirmed that pen drawing allowed the formation of films with appropriate thickness and uniformity without any obstacles.

Figure R1 (Figure 4 in the manuscript). Surface morphology of each layer formed by pen drawing. AFM topographic images, height section analyses, and three-dimensional images of PH1000 (a), PEDOT:PSS (b), PVK (c), QDs for G (d), ZnO (e), and Ag NWs (f).

Figure R2 (Figure S1 in the Supplementary Information). Classical Landau–Levich model. When the pen drawing speed increases, the film thickness of each layer constituting the pen drawing display structure also increases.

In addition, OLED devices were fabricated on a plastic substrate by the pen drawing method using red/green/blue doped-PVK solutions in which 1 wt% PVK (host material) and bis(2-methylidibenzo[*f,h*]quinoxaline)(acetylacetonate)iridium(III) ($\text{Ir}(\text{MDQ})_2(\text{acac})$) (red dopant)/Tris[2-phenylpyridinato- C^2, N]iridium(III) ($\text{Ir}(\text{ppy})_3$) (green dopant)/bis[2-(4,6-difluorophenyl)pyridinato- C^2, N](picolinato)iridium(III) (FIrpic) (blue dopant) were respectively dissolved in chlorobenzene at a weight ratio of 10:1. Figure R3 shows the emission image of OLEDs composed of two electrodes (ITO and Aluminum) formed by sputter method and three organic layers (PEDOT:PSS, red/green/blue doped-PVK solutions, and bathophenanthroline (BPhen)) formed by the pen drawing method. Note that the drawing speed, substrate temperature, and drawing cycle of the PEDOT:PSS layer were 20 mm/s, 60 °C, and 1 half-cycle, respectively. In case of the RGB doped-PVK and BPhen layers, the drawing speed, substrate temperature, and drawing cycle were 20 mm/s, 20 °C, and 1 half-cycle, respectively. In addition, using RGB-doped PVK solutions as an EML layer, the free-style multi-color texts in a poetry paragraph are handwritten on a plastic substrate. (Figure R3) As a result, it was confirmed that a polymer-based solution can be used as an EML to produce a display using pen drawing.

Figure R3. Handwriting display images produced using the pen drawing method with an EML layer consisting of PVK as a host material, Ir(MDQ)₂(acac) as a red dopant, Ir(ppy)₃ as a green dopant, and FIrpic as a blue dopant.

Comments 3. Were the pens used empty of ink or were they emptied and washed prior use?

Response: Each felt-tip pen used in this study was emptied of any existing ink, with residual ink removed by sonicating the pen in IPA for 1 day. The solvent containing the felt-tip pen was changed frequently until it became colorless. Subsequently, the pen was placed in DI water and ultrasonicated for 1 day. Then, it was removed and dried naturally for 1 day.

To clarify this point, we have added an additional explanation on page 18 (highlighted in blue), as follows:

On page 18, “Each felt-tip pen was emptied of all existing ink and sonicated for 24 h in IPA to remove residual ink; the solvent containing the pen was changed frequently until it became colorless. Subsequently, the felt-tip pen was sonicated in DI water for 24 h and then dried naturally for 24 h.”

Reviewer #2 (Remarks to the Author):

The paper describes the first hand-drawn, color, electroluminescent, quantum dot (QD) display. Prior work has reduced the complexity of QD display fabrication from a cleanroom process, to a printed process (especially Ref. 4). This work further reduces the barrier to entry for QD display fabrication by replacing printing with a hand-drawn process while preserving and extending the ability to fabricate on a wide variety of stiff and flexible substrates. The results are good, considering the fabrication method. The artifacts adequately establish the feasibility of hand-drawn quantum dot displays, especially Figure 7.

Comments 1. Abstract: What is the significance of ‘boardless’ in this context? How is this display boardless when it requires a substrate (e.g. glass and plastic per Figure 6)? As a primary motivation, diversifying display form factors seems weak without an example to separate it from the previous printing work of Ref 4. What are ‘frame’ manufacturing techniques?

Response: We appreciate the reviewer’s comment. An advantage of the pen drawing display is that the entire complex fabrication process for the display is encapsulated in a pen. Therefore, it is possible to implement a display having various color patterns by using a display pen only, without requiring conventional complex display fabrication processes such as deposition, alignment, patterning, and transferring. The main goal of the pen drawing display is to increase the applicability of displays through a differentiated fabrication method, not to replace conventional commercialized large displays (LCD, OLED, and QD-LED).

In this sense, “boardless” was used as an expression to emphasize the advantage of the pen drawing display for facilitating the formation of free patterns by hand without the constraints of substrate shape or size. Furthermore, “frame” was used an expression to show the contrast with formal fabrication technologies of regular displays, which are deposited in the form of pixels on rectangular substrates.

In the case of Ref. 4, the transfer method using PDMS can be used to fabricate displays on flat, curved, and convoluted surfaces, similar to the pen drawing method. However, with respect to the fabrication process, the pen drawing method allows displays to be fabricated by using a pen only, whereas the conventional method requires additional deposition, transferring, patterning, and aligning processes using PDMS.

The differentiation and applicability of the pen drawing display are as follows. (i) It is possible to fabricate a display by only using a pen. Thus, displays can be produced easily and rapidly at a desired location. (ii) There is no limitation on the substrate (e.g., paper, plastic, and glass) or the surface morphology of the product (curved, large area, or complex structure); thus, it is easy to produce indoor and outdoor displays/illuminations/signboards for building walls, windows, department store pillars, offices, and vehicles as well as for places where it is difficult to install conventional displays, such as mountains and islands, or places where signboards must be replaced frequently. As a typical example, a user could easily display an advertisement consisting of various text characters or image shapes without being affected by the size or curvature of the glass window of a store.

To clarify this point, we have added an additional explanation on page 2 (highlighted in blue), as follows:

On page 2, “This study proposes a pen drawing display technology that can realize a boardless display in any form based on the user’s preferences, without the usual restrictions of conventional frame manufacturing techniques. An advantage of the pen drawing method is that the entire complex fabrication process for the display is encapsulated in a pen. Therefore, it is possible to implement a display having various color patterns by using a display pen only, without requiring conventional complex display fabrication processes such as deposition, alignment, patterning, and transferring.”

Comments 2. Introduction: How do flexible, transparent and holographic displays address the problem of fixed rectangular image shape?

Response: We appreciate the reviewer’s comment. In the corresponding sentence, it seems that the intended meaning was unclear because of the use of the words “flexible” and “transparent”. To clarify this point in the Introduction, we have provided an additional explanation on page 3 (highlighted in blue), as follows:

“However, as current displays (such as computer monitors, smartphone displays, and navigation displays) are manufactured with a fixed rectangular shape, it is impossible to manipulate them to reflect the various design ideas of display users. Studies have been actively conducted on display fabrication methods based on printed assemblies of LEDs, QD (EML material) transfer, intaglio transfer, screen printing, and jet printing to overcome the constraints of rectangular-shaped displays.¹⁻⁷”

1 Park, S.-I. *et al.* Printed assemblies of inorganic light-emitting diodes for deformable and semitransparent displays. *Science* **325**, 977-981 (2009).

2 Kim, T.-H. *et al.* Full-colour quantum dot displays fabricated by transfer printing. *Nature Photonics* **5**, 176-182 (2011).

3 Choi, M. K. *et al.* Wearable red–green–blue quantum dot light-emitting diode array using high-resolution intaglio transfer printing. *Nature Communications* **6**, 7149 (2015).

4 Zheng, H. *et al.* All-solution processed polymer light-emitting diode displays. *Nature Communications* **4**, 1971 (2013).

5 Pardo, D. A., Jabbour, G. E. & Peyghambarian, N. Application of screen printing in the fabrication of organic light-emitting devices. *Advanced Materials* **12**, 1249-1252 (2000).

6 Kim, B. H. *et al.* High-resolution patterns of quantum dots formed by electrohydrodynamic jet printing for light-emitting diodes. *Nano Letters* **15**, 969-973 (2015).

7 Olberding, S., Wessely, M. & Steimle, J. In *Proceedings of the 27th Annual ACM Symposium on User Interface Software and Technology*, 281-290 (ACM, Honolulu, Hawaii, USA, 2014).

Comments 3. Page 3. The claim that this is the first fully pen drawing display is both significant and novel for QD displays. The creation of a display that is constructed entirely on a bench and on a wide variety of substrates has been previously reported both for QDs and on other electroluminescent platforms. EL (TFEL) is an example of a display technology that can be screen-printed, with multiple colors, on a variety of materials (wood, plastic, leather). See reference below: Olberding, Simon, Michael Wessely, and Jürgen Steimle. "PrintScreen: fabricating highly customizable thin-film touch-displays." Proceedings

of the 27th annual ACM symposium on User interface software and technology. ACM, 2014. Perhaps the authors should acknowledge TFEL and screen-printed displays. QDs may have a number of significant advantages (lower drive voltage, higher efficiency) but it seems appropriate to acknowledge these displays as they have shared many of the goals set out in this paper and they are present in many of the target applications mentioned in the introduction.

Response: We appreciate the reviewer's comment. The introduction has been revised in accordance with the reviewers' suggestions. The suggested reference has been also added.

“However, as current displays (such as computer monitors, smartphone displays, and navigation displays) are manufactured with a fixed rectangular shape, it is impossible to manipulate them to reflect the various design ideas of display users. Studies have been actively conducted on display fabrication methods based on printed assemblies of LEDs, QD (EML material) transfer, intaglio transfer, screen printing, and jet printing to overcome the constraints of rectangular-shaped displays.¹⁻⁷”

7 Olberding, S., Wessely, M. & Steimle, J. In *Proceedings of the 27th Annual ACM Symposium on User Interface Software and Technology*, 281-290 (ACM, Honolulu, Hawaii, USA, 2014).

Comments 4. Page 5. How sensitive are the display characteristics to the parameters listed here such as pen speed, material thickness, and uniformity? Can we reasonably expect a hand-drawn display to produce results similar to the carefully controlled machine-drawn results described? Was the work in Figure 7 the result of multiple attempts with these parameters varied? I feel that this may warrant some discussion as it is the contribution of this paper which diverges most significantly from previous work.

Response: In general, the luminance and efficiency characteristics of the QD-LED device are very sensitive to the thickness of each layer (PH1000, PEDOT:PSS, PVK, QDs, ZnO NPs, and Ag NWs). Therefore, to obtain a value as similar as possible to the optimized thickness of a QD-LED fabricated using the conventional spin-coating method, three drawing conditions (drawing speed, number of drawing cycles, and substrate temperature) were controlled. In detail, by controlling the drawing speed, the film thickness was controlled, and by controlling the drawing cycle and substrate temperature, the uniformity of the film was controlled.

The mechanical drawing method was used for examining the change in film thickness according to the drawing speed, which was the most important variable in the pen drawing method. Using the drawing speed conditions derived here, the film thickness was controlled during the hand drawing method. For an identical length, when hand drawing and mechanical drawing were performed at similar drawing speeds, the obtained film thicknesses were similar.

The handwriting images shown in Figs. 6 and 7 are the results of fabricating pen-drawing devices on glass, plastic, and paper by using a different drawing speed, drawing cycles, and substrate temperature for each layer, as determined by the results obtained up to Fig. 5. In other words, the three drawing conditions (drawing speed, number of drawing cycles, and substrate temperature) of the pen drawing display were in a fixed state (Table R1).

Table R1. Hand-drawing conditions and thickness of each layer of the pen drawing display formed on glass, plastic, and paper.

Material	Drawing conditions			Layer thickness (nm)
	Drawing speed (mm/s)	Drawing cycles	Substrate temperature (°C)	
PH1000 (anode)	20	5 cycles	60	~100
PEDOT:PSS (HIL)	20	1 half-cycle	60	~40
PVK (HTL)	20	1 half-cycle	20	~50
QDs (EML)	20	1 half-cycle	20	~30
ZnO NPs (ETL)	20	1 half-cycle	20	~60
Ag NWs (cathode)	20	3 half-cycles	60	~600

Comments 5. Page 14. The first sentence of the last paragraph is not very precise. What is meant by the words, ‘actual device.’

Response: We appreciate the reviewer’s comment. The expression “actual device” was used to indicate that all the QD layers from the anode/cathode electrodes were implemented on paper by using the pen drawing method. However, as the intended meaning was ambiguous, this expression has been revised as follows:

On page 15, “Figure 7 shows the process for producing a pen drawing display on paper and light-emitting RGB images of the fabricated device.”

Comments 6. References: The references could include TFEL as noted above.

Response: At the request of the reviewer, we have made additions and changes to the references in the manuscript.

On page 20, “1 Park, S.-I. *et al.* Printed assemblies of inorganic light-emitting diodes for deformable and semitransparent displays. *Science* **325**, 977-981 (2009).

2 Kim, T.-H. *et al.* Full-colour quantum dot displays fabricated by transfer printing. *Nature Photonics* **5**, 176-182 (2011).

3 Choi, M. K. *et al.* Wearable red–green–blue quantum dot light-emitting diode array using high-resolution intaglio transfer printing. *Nature Communications* **6**, 7149 (2015).

4 Zheng, H. *et al.* All-solution processed polymer light-emitting diode displays. *Nature Communications* **4**, 1971 (2013).

5 Pardo, D. A., Jabbour, G. E. & Peyghambarian, N. Application of screen printing in the fabrication of organic light-emitting devices. *Advanced Materials* **12**, 1249-1252 (2000).

6 Kim, B. H. *et al.* High-resolution patterns of quantum dots formed by electrohydrodynamic jet printing for light-emitting diodes. *Nano Letters* **15**, 969-973 (2015).

7 Olberding, S., Wessely, M. & Steimle, J. In *Proceedings of the 27th Annual ACM Symposium on User Interface Software and Technology*, 281-290 (ACM, Honolulu, Hawaii, USA, 2014).”

Comments 7. Figures: The figures are effective artifacts and the photos appear to be free of inappropriate modification.

Response: We appreciate the reviewer’s positive comment. Indeed, all the photo images in the present manuscript are shown without processing. As the photo image of step 4 in Fig. 7 may be difficult to understand for general readers, the corresponding drawing processes have been separated and shown in detail as step 4-1 to step 4-3.

Figure R4 (Figure 7 in the manuscript). Handwriting images using the pen drawing method. Photo images of “Display” handwritten on paper, showing the light-emitting ability, mechanical flexibility, and tearing properties.

Reviewer #3 (Remarks to the Author):

This study offers an interesting inexpensive approach to customizing the formation of simple devices using pen drawing. The process and concept are minimally described - a number of general statements and claims are made that need justification and in some cases, an indication of values and the variance observed or expected.

Comments 1. Lines 79-81: What is the range of line widths that can be readily generated? Does this require pens with different tips? What is the variance or reproducibility in line width for each tip? How does this affect device properties such as luminance of the QD-LEDs?

Response: We appreciate the reviewer's comment. To investigate the relationship between the width of the felt-tip pen and the line width formed on a substrate, pen drawing of a PH1000 film was performed using felt-tip pens with various widths, which dictated the line shape of the obtained film (Fig. R5). In addition to the felt-tip pen used for bold text (width, 4.0 mm; DONG-A, Twinliner SOFT) in this study, felt-tip pens with widths of 7.4, 2.6, and 1.0 mm were used. The PH1000 films were formed on a plastic substrate using the following drawing conditions: 5 cycles of pen drawing, a drawing speed of 20 mm/s, and a substrate temperature of 60 °C. Here, the line widths of the layers formed by pen drawing were 7.5, 4.0, 2.6, and 1.2 mm, which were almost identical to the widths of the felt-tip pens used (7.4, 4.0, 2.6, and 1.0 mm, respectively). Therefore, a user can form a desired line width on a display by drawing the required area using either multiple lines with a thin felt-tip or a single line with a thick felt-tip.

Furthermore, the layer roughness determined through AFM measurements (R_{rms} values of 1.62, 1.61, 1.82, and 2.26 nm for felt-tip pen widths of 7.4, 4.0, 2.6, and 1.0 mm, respectively) showed similar results (Fig. R5). In this study, the layer-formation images and transmittance and sheet resistance results corresponded to the averages and standard deviations for 10 repeated measurements. Accordingly, it was ascertained that reliable layer formation and QD-LED fabrication could be facilitated through pen drawing.

Figure R5 (*Figure S5 in the Supplementary Information*). Line images and surface roughness properties of the PH1000 layer formed by felt-tip pens with different pen tip widths: 7.4 (a), 4.0 (b), 2.6 (c), and 1.0 (d) mm.

To clarify this point, we provided an additional explanation in the Supplementary Information on page S–8 (highlighted in blue) as follows:

On page S–8, “To investigate the relationship between the width of the felt-tip pen and the line width formed on a substrate, pen drawing of PH1000 films was performed using felt-tip pens with various widths, which dictated the line shape of the obtained film (Fig. S5). In addition to the felt-tip pen used for bold text (width, 4.0 mm; DONG-A, Twinliner SOFT) in this study, felt-tip pens with widths of 7.4, 2.6, and 1.0 mm were used. The PH1000 films were formed on a plastic substrate using the following drawing conditions: 5 cycles of pen drawing, a drawing speed of 20 mm/s, and a substrate temperature of 60 °C. Here, the line widths of the layers formed by pen drawing were 7.5, 4.0, 2.7, and 1.2 mm, which were almost identical to the widths of the felt-tip pens used (7.4, 4.0, 2.6, and 1.0 mm, respectively). Therefore, a user can form a desired line width on a display by drawing the required area using either multiple lines with a thin felt-tip or a single line with a thick felt-tip. Furthermore, the layer roughness determined through AFM measurements (R_{rms} values of 1.62, 1.61, 1.82, and 2.26 nm for felt-tip pen widths of 7.4, 4.0, 2.6, and 1.0 mm, respectively) show similar results (Fig. S5). In this study, the layer-formation images and transmittance and sheet resistance results correspond to the averages and standard deviations for 10 repeated measurements. Accordingly, it was ascertained that reliable layer formation and QD-LED fabrication could be facilitated through pen drawing.”

Comments 2. Lines 99-102: What is a "suitable temperature"? Film properties will vary with heating temperature and probably heating time, so this statement gives no useful information.

Response: We appreciate the reviewer's comment. As the reviewer pointed out, the expression "suitable temperature" is not appropriate for providing accurate information. Among the solutions used in this study, the Ag NW solution, which was the cathode, was dispersed in IPA, whereas for the ZnO solution, which was the ETL, the solvent was ethanol. Therefore, when pen drawing was performed with the Ag NWs solution, if the solvent (IPA) is not evaporated quickly, there is a possibility that ZnO in the previous layer may be dissolved in IPA, thus damaging the layer. Therefore, although IPA is highly volatile, pen drawing with Ag NWs was performed at a substrate temperature of 60 °C to evaporate it more quickly.

To clarify this, the process temperature conditions have been provided.

On page 6, "In addition, the evaporation speed of the solvent remaining on the thin film formed by the pen drawing was controlled by applying a temperature of 60 °C to the substrate so as not to damage the lower layer in the process of forming a specific layer using the pen drawing method."

Comments 3. Lines 151-154: The process apparently balances convective flow and capillary flow. How controllable and reproducible is this effect? The terms uniformity (line 155) and uniform (line 166) are used, but these terms have no meaning. How reproducible are the values of sheet resistance and transparency? Does this depend upon the air flow rate, humidity, room temperature, etc.? What variance is observed?

Response: We appreciate the reviewer's comment. The effect of convection flow and capillary flow inside the solution in the process of pen drawing can be controlled through the drawing speed and substrate temperature. If the coating speed is similar to or lower than the evaporation speed of the solvent (< 0.1 mm/s), the solution is dried immediately after pen drawing. Therefore, in this case, the effect of convective flow and capillary flow in the meniscus solution can be ignored during pen drawing. In contrast, if the coating speed for pen drawing is faster than the evaporation speed of the solvent, a wet film is formed outside the meniscus in accordance with the classical Landau–Levich regime. According to the classical Landau–Levich regime, owing to convective flow and capillary flow, as the solvent evaporates, the solute moves to the side of the deposition contact line where the concentration has increased, and the thickness of the film increases. Therefore, as the drawing speed increases, capillary flow increases, and consequently, the thickness of the film increases. In general, convective flow (Marangoni flow) in the meniscus solution is driven by the surface tension gradient, which is induced by the distribution of the local temperature and/or the concentration along the meniscus surface [R1]. Increasing the substrate temperature increases the rate of solvent evaporation, resulting in a concentration gradient of the solute in the deposition contact line. Therefore, convective flow in the meniscus area in the direction of the deposition contact line can be increased to increase the thickness in the deposition contact line [R2].

PH1000 for the anode has a particular core-shell structure composed of hydrophobic PEDOT molecule cores surrounded by shells with hydrophilic PSS chains. As water (the solvent of PEDOT:PSS) has low volatility at room temperature (20 °C), a wet PH1000 film is formed by pen drawing when slowly dried. Therefore, when the wet film of PH1000 was slowly dried and the amount of water was reduced, the hydrophobic PEDOT molecule cores were gradually exposed on the surface. Consequently, owing to hydrophilic-hydrophobic repulsion on the hydrophilic substrate, a problem occurred in that the PEDOT molecules aggregated at the center of the wet PH1000 film. When this occurs, because an anode layer of uniform thickness and roughness with the desired area is not formed, uniform light emission from the LED device becomes impossible. To solve this problem, we investigated the conditions for forming an anode layer of uniform thickness and roughness by controlling the substrate temperature and drawing cycle (Figs. R6 and R7). In this study, the layer-formation images and transmittance and sheet resistance results corresponded to the averages and standard deviations for 10 repeated measurements. Thus, it is ascertained that the formation of a reliable layer and the fabrication of QD-LEDs through pen drawing are possible.

Figure R6 (Figure S2 in the Supplementary Information). Optical and electrical properties of the PH1000 layer according to the substrate temperature at a drawing speed 20 mm/s. Sheet resistance (a) and optical transmittance (b) at the center and edge of the PH1000 film at substrate temperatures of 20, 40, 60, and 80 °C in the case of 1 drawing cycle.

Figure R7 (Figure S3 in the Supplementary Information). Optical and electrical properties of the PH1000 layer according to the number of drawing cycles at a drawing speed of 20 mm/s and a substrate temperature of 20 °C.

substrate temperature of 60 °C. Sheet resistance (a) and optical transmittance (b) characteristics of the PH1000 layer measured at the center and edge. c. Optical images of the PH1000 layer according to the number of drawing cycles.

In this study, the pen drawing process was performed under a temperature and relative humidity of 20 °C and 30%, respectively. During the pen drawing display fabrication process, the temperature and relative humidity were one of the most important environment conditions, but the effect of the air flow rate was ignored. To verify if the pen drawing conditions changed with changes in the temperature and relative humidity, the pen drawing process was performed to form a PH1000 layer after increasing the temperature/relative humidity to 26 °C/45% and 26 °C/85%. Here, the drawing conditions were 5 cycles of pen drawing, a drawing speed of 20 mm/s, and a substrate temperature of 60 °C. It was confirmed that the PH1000 layer formed by pen drawing was uniformly coated on the optical image without solution aggregation. In addition, the surface roughnesses determined by AFM measurements were 1.61 (20 °C/30%), 1.67 (26 °C/45%), and 1.80 nm (26 °C/85%); the difference when the temperature and humidity were changed was not significant. Consequently, there was an insignificant difference between the layers formed by pen drawing. Thus, it was confirmed that the membrane of the layer formed by the pen drawing was maintained consistently, even when the external environment changed.

Figure R8. Line images and surface roughness properties of the PH1000 layers formed at (a) 20 °C/30%, (b) 26 °C/45%, and (c) 26 °C/85%.

[R1] Parsa, M., Harmand, S. & Sefiane, K. Mechanisms of pattern formation from dried sessile drops. *Advances in Colloid and Interface Science* **254**, 22-47 (2018).

[R2] Kim, J.-H., Park, S.-B., Kim, J. H. & Zin, W.-C. Polymer transports inside evaporating water droplets at various substrate temperatures. *The Journal of Physical Chemistry C* **115**, 15375-15383 (2011).

Comments 4. Lines 283-285: What does "somewhat lower" mean?

Response: This description was intended to indicate that the light-emitting characteristics of the QD-LED fabricated with the pen drawing method are inferior to those of the QD-LED device fabricated using the spin-coating method, the most common process for fabricating conventional QD-LEDs. The poor luminance and efficiency of the fabricated devices were mainly due to the high resistivity of the electrodes, relatively poor contact resistance, poor transmittance, and the lack of reflection from the transparent metallic cathode. Obtaining a high luminance with a low turn-on voltage is dependent on the sheet resistance of the electrodes. Compared with the sheet resistances of $\sim 8.7 \Omega/\text{sq}$ for ITO and ~ 1.3

Ω/sq for Al, the sheet resistances of the pen-drawn PH1000 ($\sim 30 \Omega/\text{sq}$) and Ag NWs ($\sim 110 \Omega/\text{sq}$) as anode and cathode, respectively, were considerably higher. This difference in the resistance characteristics seems to have significantly affected the luminance and efficiency. Furthermore, the characteristics also depend on whether the luminance is measured at the anode or cathode of the produced device. For conventional light-emitting devices, the luminance is measured in the direction of the transparent anode (ITO). Thus, the measured luminescence and efficiency increase because the light emitted from the LED is reflected by the opaque and glassy metallic cathode (Al), and it progresses in the anode direction. However, for the LED produced by pen drawing, there is no opaque and glassy material such as Al in the lower film, and only some of the light emitted by the Ag NW layer as the cathode with 70% transparency is reflected.

To clarify this point, we modified the explanation on page 14 (highlighted in blue), as follows:

On page 14, “As a result, the measured luminance and efficiency of the pen drawing display in which the anode and cathode were formed by the pen drawing method were lower than those of the pen drawing display with forming the anode and cathode by the conventional sputtering process.”

Comments 5. Lines 362-366: After depositing 300 nm of SiO_2 onto paper, the surface is essentially that of glass. What is the purpose of using paper substrates in this situation (I presume it is a cost issue, but the deposition adds cost also). Does the paper maintain its flexibility after deposition? Does the glass crack upon bending? Is the substrate curved after film deposition due to differences in thermal expansion coefficient?

Response: The reason for coating a SiO_2 film on the paper substrate was mentioned on page 15 of the manuscript submitted previously.

On page 15, “Furthermore, a 300 nm SiO_2 thin film was deposited on the paper substrate to prevent the solution from permeating into the cellulose fibers and making the thin film thinner or irregular.”

As shown in Fig. S9, the flexibility of the substrate was maintained even after coating the SiO_2 film. As the thickness of the SiO_2 film was 300 nm, which was very thin compared with the thickness of the paper (0.25 mm), the flexibility of the paper was excellent, and no noticeable bending of the paper substrate occurred (Fig. R9). Therefore, there was no problem with proceeding with the pen drawing.

Figure R9. Photo images of conventional photo paper (left) and 300 nm thick SiO₂-deposited photo paper (right)

If the substrate is bent, cracks can occur in the SiO₂ film coated on the paper. Currently, flexible substrate-applicable organic- or inorganic-based insulators are under development [R3–R7]. Application of such materials in the future will likely solve this cracking problem.

[R3] Seo, H.-K. *et al.* Efficient flexible organic/inorganic hybrid perovskite light-emitting diodes based on graphene anode. *Advanced Materials* **29**, 1605587 (2017).

[R4] Choi, K. *et al.* Reduced water vapor transmission rate of graphene gas barrier films for flexible organic field-effect transistors. *ACS Nano* **9**, 5818-5824 (2015).

[R5] Wang, L. *et al.* Enhanced moisture barrier performance for ALD-encapsulated OLEDs by introducing an organic protective layer. *Journal of Materials Chemistry C* **5**, 4017-4024 (2017).

[R6] Chen, Z. *et al.* Low-temperature remote plasma enhanced atomic layer deposition of ZrO₂/zirconium nanolaminate film for efficient encapsulation of flexible organic light-emitting diodes. *Scientific Reports* **7**, 40061 (2017).

[R7] Jeong, E. G., Han, Y. C., Im, H.-G., Bae, B.-S. & Choi, K. C. Highly reliable hybrid nano-stratified moisture barrier for encapsulating flexible OLEDs. *Organic Electronics* **33**, 150-155 (2016).

Comments 6. Nearly all figures in the manuscript and in Supplementary Information will be very difficult to read when reduced to fit in journal columns. Text and numbers must be MUCH larger, especially on Figs. 2-6 and Figs. S2-S6.

Response: We appreciate the reviewer's comment. The numbers and text in the figures pointed out by the reviewer (Figs. 2–6 and Figs. S2–S6) were enlarged to improve readability. Please refer to the revised figures in the manuscript and the supplementary information.

REVIEWERS' COMMENTS:

Reviewer #1 (Remarks to the Author):

The manuscript from Ju et al. describes the fabrication of LEDs and displays by direct pen writing of conductive and fluorescent inks. The great novelty of this work is the demonstrated ability to write in a layer-by-layer fashion in order to generate functional devices such as displays on unconventional surfaces, which therefore provides an alternative method to more traditional and costly coating methods. The authors have described in great detail the variation in experimental conditions leading to optimum homogenous deposition of components one on top of the other. I am also satisfied that the authors were able to reply to all raised comments with convincing data and explanation. I therefore am happy to recommend the publication of the manuscript in its present format.

Reviewer #2 (Remarks to the Author):

I was pleased with the remarkably comprehensive rebuttal and I'm satisfied with the changes made.

Reviewer #3 (Remarks to the Author):

Authors have addressed my concerns for the initial version of this manuscript. It is now suitable for publication.

Response Letter

We have made revisions in response to all the specific comments, and are submitting the revised manuscript and Supplementary Information (enclosed) for further consideration for publication in *Nature Communications*.

Comments 1. 3rd PARTY MATERIAL. Please provide us with more information on the creation of Figure 1a images. Were these images and every element of these images created by you and/or your co-Authors? Were they created using any previously-created elements? If the images are from a database please supply proof of permission for its use (receipt, express permission from creator, confirmation of compatible Open Access license or Public Domain).

Response: The images including every element in Figure 1a were created by Sang-Mi Jeong who is the 1st author of this paper, and does not mimic the external image. It also does not require permission from other creators.

Comments 2. Your Competing Interests declaration should not specify solely Competing Financial Interests. Please modify your declaration accordingly. Acceptable declarations include but are not limited to the following: - "the Authors declare no competing interests"- "the Authors declare the following competing interests..."

Response: The sentence has been modified as follows. (Highlighted in blue)

On page 25, "The authors declare no completing interests."

Comments 3. Please cite Supplementary Figure 5 at least once in the main text.

Response: 'Supplementary Figure' was cited on page 12 in the main text. (Highlighted in blue)

On page 12, "To investigate the relationship between the width of the felt-tip pen and the line width formed on a substrate, pen drawing of PH1000 films was performed using felt-tip pens with various widths, which dictated the line shape of the obtained film (Supplementary Figure 5)."

Comments 4. Please remove the individual supplementary figure files from your submission. The Single Supplementary Information PDF file is all we require.

Response: We have removed the individual supplementary figure files from the upload system, and the Single Supplementary Information PDF file is uploaded.